# Material Flow in Ultrasonic Orbital Microforming

**Wojciech Presz**

Institute of Manufacturing Technologies, Warsaw University of Technology, Warsaw 02-524, Poland;
w.presz@wip.pw.edu.pl; Tel.: +48-696-844-998

**Abstract:** Ultrasonic orbital microforming—UOM—uses the broadly understood idea of orbital forging but uses very different laws of physics. The only shaping force in this process is the inertia force resulting from the acceleration in the rotary motion of the workpiece. Micro specimen blanked from cold rolled aluminum sheet metal was used in the applied UOM process. Only the upper and lower part of the sample is deformed that gives about 70% of volume. The rest—the middle part—remains undeformed. The final shape of the product is influenced by the shape of the inside of the die in which the UMO process is carried out. However, this effect is not a direct one. The product shape does not repeat the shape of the interior of the die. The preliminary experiments with modular micro-die have been performed on the way of controlling the shape of deformed micro-objects. The microstructure analysis has been done as well as micro-hardness distribution.

**Keywords:** microforming; ultrasonic; orbital forming

---

## 1. Introduction

The progressive development of miniature systems is the driving force and challenge for miniaturization of all material technologies. The metal forming technology plays among them a significant role. Its miniaturization is connected with difficulties if object dimension is smaller than 1 mm. As a result, a new branch of metal forming which deals with the production of objects that meet the above criterion was created [1,2]. The microforming was introduced as a separate technology [3]. Deviations from the developed technology rules for years associated with miniaturization are called "scale effects". They concern, in principle, all elements of the technological process. Surface layer and contact phenomena [4–6], affecting the friction [7–9], lubrication [10], and galling [11], increasing the role of preparation methods for micro-billets surfaces [12,13]. They also refer to the internal structure [14–16], which affects the quality of the surface and cracking mechanisms [17,18]. Scale effect concerns even the construction of machines [19], tooling [20,21], and tools [22–26], as well as the design of the technological process plan [27,28]. Materials and friction tests using microsamples are also created because their results differ from those obtained in macrotests [29,30]. Reducing the size of manufactured parts on one hand is a serious challenge for traditional technologies, but on the other hand, mainly by removing the energy barrier, opens the opportunity of using other unconventional techniques. Electric power and magnetic wave assistance, laser treatment, and utilization of mechanical vibrations at various frequencies during forming can be mentioned here. In the last of these methods, ultrasonic vibrations are particularly in the sphere of greatest interest [31,32]. Ultrasonic-vibration reduce forces in the ECAP [33] and micro-extrusion processes by strong reduction of friction [34] and reduce the flow stress [35] by not only temperature increase, but also the softening effect. The use of ultrasonic vibrations can also cause phase changes [36] and specific macroscopic consequences in terms of cracking [37]. In studies on the influence of ultrasonic vibrations on the course of microforming process, often used is micro-upsetting under dry friction conditions [38,39] due to geometric simplicity and relative ease for modelling. In some conditions of upsetting with the use of ultrasonic vibrations, the

formation of a specific shape of the lateral surface of the sample is reported [40–42]. The phenomenon leading to it was taken to call "anti-barrelling". The name is a reference to the phenomenon called "barrelling" [43–45], which concerns the formation of the convex side surface of the upset cylinder. It is created as a result of friction forces occurring on the contact surfaces with tools. Anti-barrelling is the formation of a concave lateral surface. This effect is not being observed in any upsetting conditions involving ultrasonic vibrations [46,47]. The causes and consequences of this phenomenon are not fully explained and are currently under investigation. One of the hypotheses is the temporary detachment of the surface of the punch from the surface of the sample [48,49]. The ultrasonic orbital microforming (UOM) process also leads to the creation of anti-barrelling shape however, it is based on completely different physical phenomena. UOM process was introduced in 2018 by Presz [50]. The punch is vibrating with the amplitude of magnitude large enough to obtain the so called "dynamic effect" that means detaching the front of the punch from the sample. At the same time, the end of the punch turns into small circles, which allows the shaped sample to be introduced into a rotary motion. This movement begins when the advancing punch touches the surface of a sample placed on the cylindrical die. The sample is lifted and stand on the edge and begins to roll around—similarly to the coin rolling on the table with the difference that the rotary motion of the sample accelerates because of actions of punch. The punch initially slides on the edge of specimen, temporary detaching from it. The edges begin to deform and their temperature increases, slippage decreases—this means that the rotational motion accelerates—which causes the "lifting" of the sample. The inertia force resulting from accelerated motion is the only shaping force, which makes this process unique. The process occurs cyclically and the workpiece is cyclically driven to rotary motion. It is a process in some way similar to the rotary forging, giving the possibility of manufacturing short objects with variable diameter. It is implemented in a micro-scale and has the potential for further miniaturization to an unpredictable degree. It is a completely new process with potential not yet recognized. Current work is a step towards understanding, controlling and possibly commercially exploiting the potential of this process.

## 2. Materials and Methods

UOM is a forming process, which, for technical reasons, can probably only be carried out as a microforming process. Since there are no known constructions of machines that would be capable of putting the macro-object in a free spin at such a high speed that it would stabilize it in space due to the gyroscopic effect. In the only so far presented version [38] UOM consists in inducing the rotational motion of the cylindrical workpiece with a flat punch vibrating with an ultrasonic frequency. The layout of the setup is shown in Figure 1.

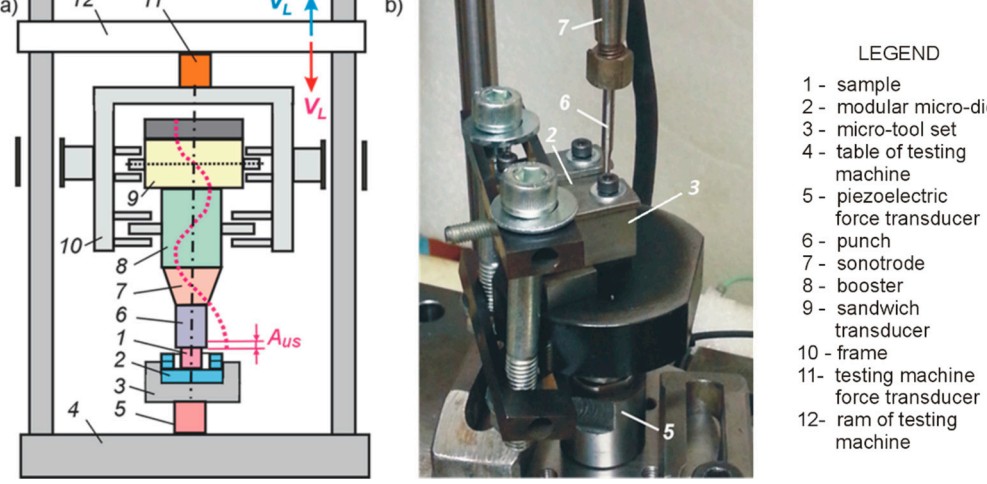

**Figure 1.** Experimental stand: (**a**) scheme of the test stand, and (**b**) close-up of working area.

The sample 1 is placed in the modular micro-die (2) in the Micro-tool set (3) standing on the force transducer (5) fixed on the table (4) of the testing machine. The sample is deformed using the punch (6), which is at the end of the sonotrode (7) of the ultrasonic system (8—booster, 9—sandwich transducer). This system is mounted in the frame (10), which through the force transducer (11) is connected to the ram (12) of the testing machine. An alternating voltage oscillating at ultrasonic frequency is applied by a power supply unit to the piezoelectric transducer. Booster and sonotrode work as half-wavelength resonators, vibrating lengthwise with standing waves at its resonant frequency. The frequency used is 20 kHz. Sonotrode acts as a displacement amplifier. On the basis of the laser displacement transducer, Keyence LK-H008, (Keyence, Osaka, Japan), the amplitude AUS of the ultrasonic vibrations on the surface of the booster and the face of the punch was determined, see Table 1.

**Table 1.** System parameters.

| Amplitude | | | Ram Velocity | |
|---|---|---|---|---|
| At Surface of Booster | At Surface of Punch Nose | Amplification of Sonotrode | Loading | Unloading |
| (µm) | (µm) | (1) | (mm/min); (m/s) | (mm/min); (m/s) |
| 2.5 | 16.0 | 6.3 | 0.2; $3.3 \times 10^{-6}$ | 0.02; $3.3 \times 10^{-7}$ |

Tools included in the test are shown in Figure 2. The micro-die used in the research has a modular structure. It consists of plates with precision holes drilled with the EDM, which can be combined in various configurations. Three versions of it were used, as shown in Figure 2: 0.2-mm thick plates designated as: C, F, and J were used. In each of presented experiments, two modules were used, matching them as follows: CC, FC, JC.

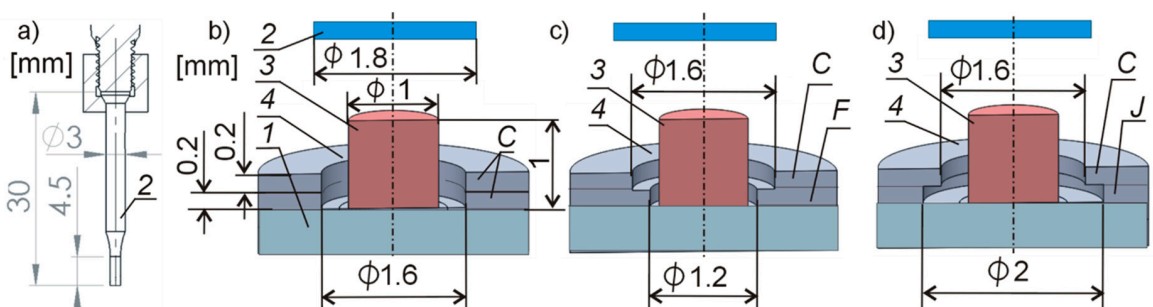

**Figure 2.** Tools included in the tests: (**a**) punch, (**b**) die CC: specimen "S3", (**c**) die FC: specimen "S4", and (**d**) die JC: specimen "S5".

The samples used for testing, see Figure 3, are blank with reduced clearance from cold rolled, 1-mm thick aluminum sheet. Diameter of samples $d = 1^{0}_{-0.02}$ mm. Samples after this process have a slightly convex upper surface, 1, and a small and very thin burr, 2, which before the process was manually removed. On some parts of the circumference, there is also a fracture zone, 3. The structure of the preforms is a modified structure after cold rolling process, see Figure 3b. Modification of the structure is associated with the zone of plastic deformation in the process of plastic blanking. In consequence there is a hardness distribution shown in Figure 11a. The plasticity characteristics of the material in the annealed state are represented by the stress strain curves given in Figure 3, the CR curve is drawn on the basis of experimentally determined equation of the two-parameters form (1). Experimental sites included only the range of $\varepsilon = (0, 0.3)$. The values of constants C and n are given in the Table 2.

$$\sigma_p = C \cdot \varepsilon^n \tag{1}$$

**Table 2.** Mechanical properties of specimen materials (cold rolled—*CR*).

| *E* (GPa) | $R_e$ (MPa) | *ν* (1) | *C* (MPa) | *N* (1) |
|-----------|-------------|---------|-----------|---------|
| 70        | 135         | 0.32    | 170       | 0.05    |

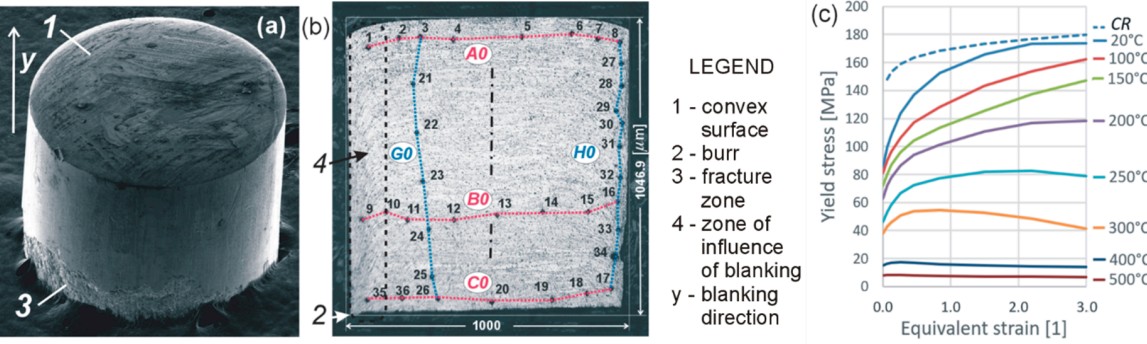

**Figure 3.** (**a**) SEM of specimen "S0": (**b**) Cross-section with marked micro-hardness, further described in text; and (**c**) stress strain curves of used material in the mild state and at the state of experiment, CR.

## 3. Results

During the processes, forces were recorded using a strain gauge dynamometer of a testing machine, Figures 1–11 and an additional piezoelectric dynamometer, Figures 1–5. The recorded runs of process forces are shown in Figure 4.

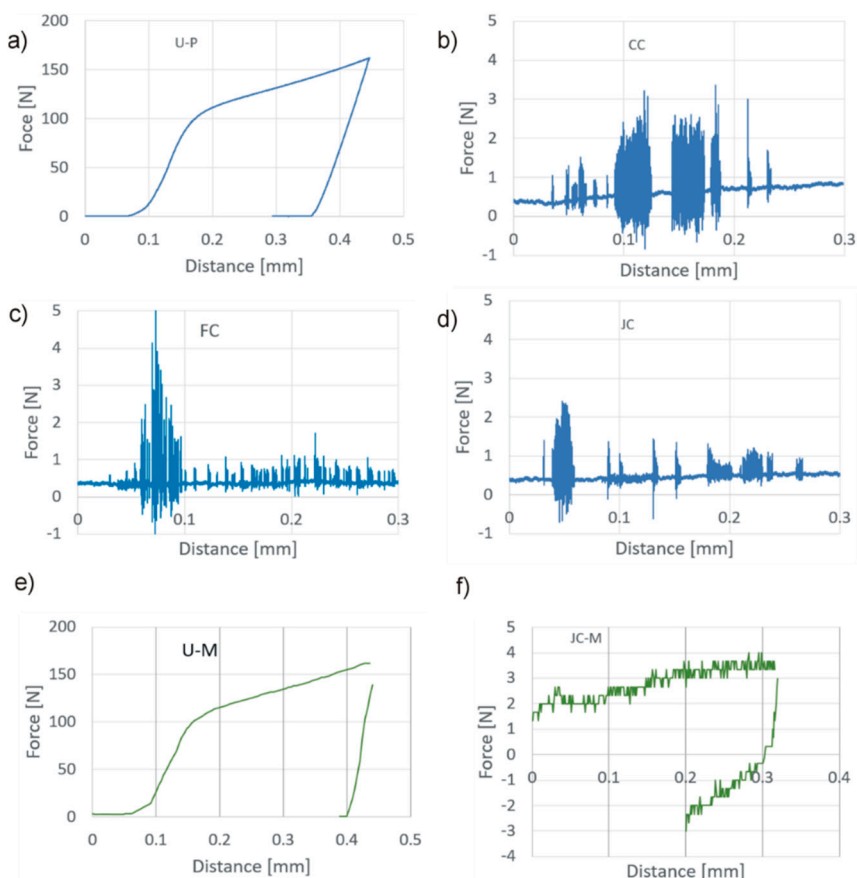

**Figure 4.** Process forces recorded by the auxiliary dynamometer (**a**)–(**d**) and by the testing machine dynamometer (**e**,**f**): (**a**) free upsetting—S1, (**b**) process CC—S3, (**c**) process FC—S4, (**d**) process JC—S5, (**e**) free upsetting—S1, and (**f**) process JC—S5.

In Figure 5 are reported images and outlines of the samples after upsetting and UOM. The outlines were obtained on the basis of microscopic photographs taken from the side.

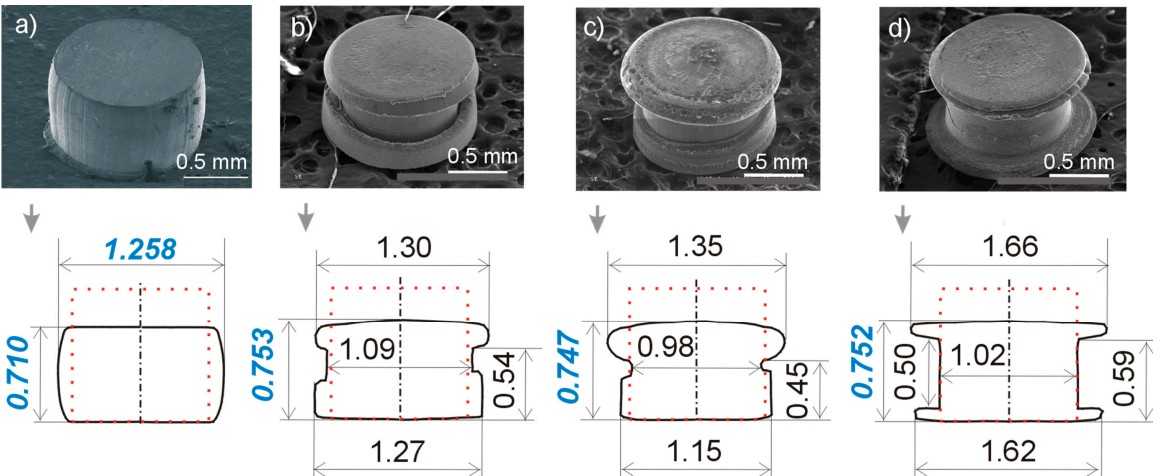

**Figure 5.** Microscopic (SEM) photos of deformed samples and their outlines drawn on the base of side views: (**a**) free expansion without vibrations—specimen S1, (**b**) UOM with a die CC—specimen S3, (**c**) UOM with a die FC—specimen S4, and (**d**) UOM with a die JC—specimen S5.

The metallographic cross-section of the samples are shown in Figure 6. On the basis of these, it was found that the samples in the middle part do not undergo deformation, and only their upper and lower parts are deformed. The cross-sections allowed to build 3D models and estimate the size of equivalent strains. In the model construction, the dimensions were determined based on the cross-sections from Figure 6 based on the height of the samples, which were measured with a micrometer.

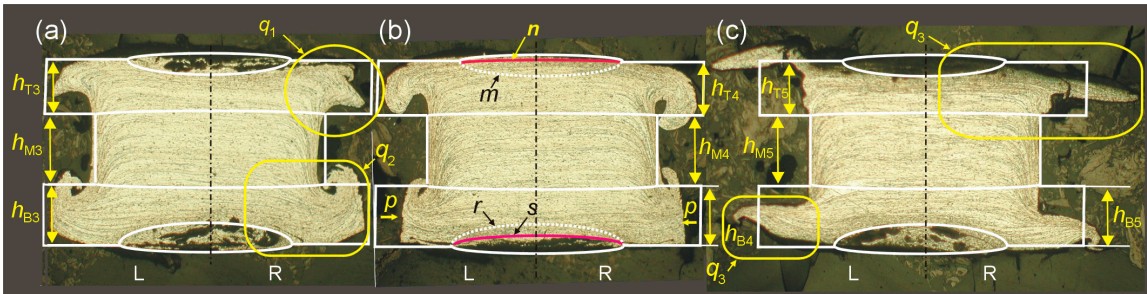

**Figure 6.** The metallographic cross-sections of the samples: (**a**) sample S3; (**b**) sample S4; and (**c**) sample S5.

The methodology for determining the average equivalent strain in deformed areas is shown in Figure 7a–d. At the beginning, the values $h^{(i)}_{T,M,B}$, $i = 3, 4, 5$ (top, middle, bottom) were determined and the half-section boundary was determined for each of the samples. Assuming the symmetry of shapes, 3D axially symmetric models of each of the specimens were prepared. Then the models were divided into three parts according to the determined $h^{(i)}_{T,M,B}$ and their volumes $V^{(i)}_{T,M,B}$ were calculated as well as the dimensions of cylinders $c^{(i)}_{T,B,n}$ and $c^{(i)}_{T,B,s}$. Assuming that their volumes are equal to the volumes $V^{(i)}_{T,M}$ and their dimensions meet (2) and (3) respectively, the values $h^{(i)}_{T,M,B,n}$ and $d^{(i)}_{T,M,B,n}$ were calculated according to (4) and (5).

$$d^{(i)}_{T,B,n} = d_0 \tag{2}$$

$$h^{(i)}_{T,B,s} = h^{(i)}_{T,B} \tag{3}$$

$$h^{(i)}_{T,B,n} = 4 \cdot \frac{V^{(i)}_{T,B}}{\pi \cdot d_0^2} \tag{4}$$

$$d^{(i)}_{T,B,n} = 2 \cdot \sqrt{\frac{V^{(i)}_{T,B,}}{\pi \cdot h^{(i)}_{T,B}}} \tag{5}$$

where all used quantities are defined in Figure 7.

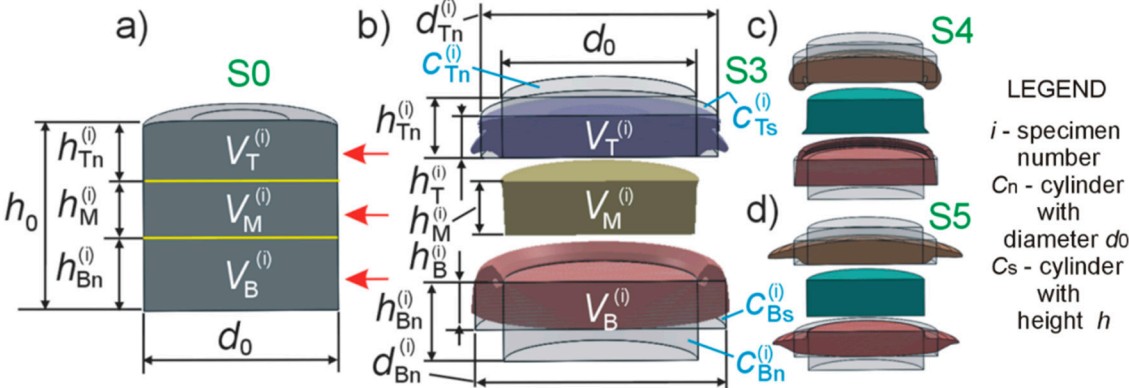

**Figure 7.** (**a**)–(**d**) The methodology for determining the average equivalent strain.

Mean equivalent strain in the upper and lower areas was calculated assuming that it is equal to the modulus of the axial deformation component according to Equation (6):

$$\overline{\varepsilon}^{(i)}_{T,B} = \left| 2 \cdot ln \frac{d_0}{d^{(i)}_{B,T,n}} \right| = \left| ln \frac{h^{(i)}_{T,B}}{h^{(i)}_{T,B}} \right| \tag{6}$$

In the sample S5 local strains were estimated according to Figure 8 and Equation (7):

$$\varepsilon^{(5)}_{T,B} = \left| ln \frac{h^{(5),\,j}_{T,B}}{h^{(5),\,j}_{T,B,n}} \right|, \ at \ r^{(5),\,j}_{T,B} \tag{7}$$

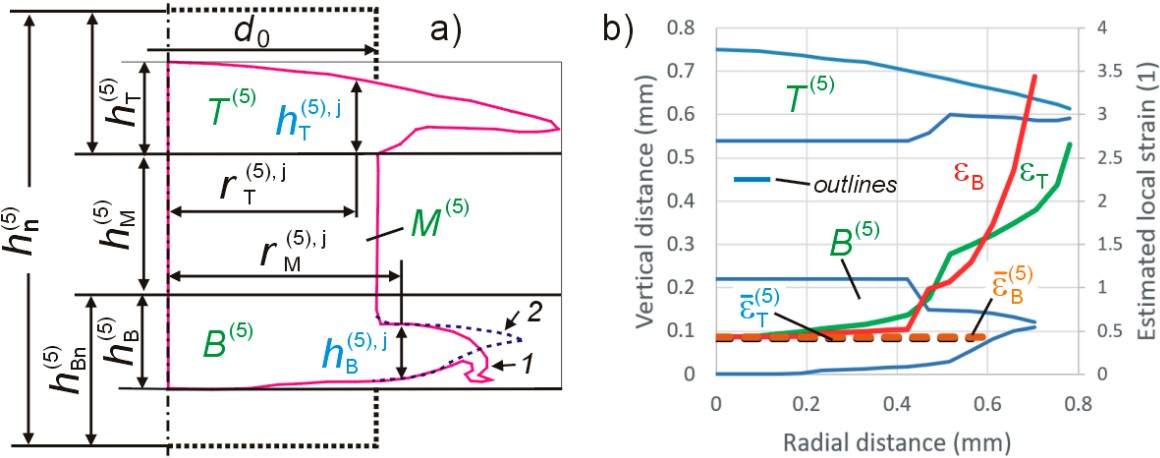

**Figure 8.** (**a**) The method of estimating local equivalent strain in specimen S5; and (**b**) mean and local equivalent strain distribution in the specimen S5.

The determined values for samples S3–S5 are collected in Tables 3 and 4. The strains distributions according to Equation (6) are shown in Figure 8b.

**Table 3.** Calculated values for *V* and *h* of specimens S3–S5.

| Specimen | $V$ (mm$^3$) | | | | $h$ (mm) | | | |
|---|---|---|---|---|---|---|---|---|
| | All | T | M | B | All | T | M | B |
| S3 | 0.783 | 0.25 | 0.206 | 0.327 | 0.752 | 0.22 | 0.283 | 0.25 |
| S4 | 0.773 | 0.262 | 0.238 | 0.273 | 0.747 | 0.223 | 0.297 | 0.227 |
| S5 | 0.781 | 0.253 | 0.262 | 0.266 | 0.752 | 0.212 | 0.32 | 0.22 |

**Table 4.** Calculated values for $h_n$, $d_s$ and $\varepsilon$ of specimens S3–S5.

| Specimen | $h_n$ (mm) | | | $d_{s.}$ (mm) | | | $\varepsilon$ (1) | | |
|---|---|---|---|---|---|---|---|---|---|
| | T | M | B | T | M | B | T | M | B |
| S3 | 0.318 | 0.263 | 0.417 | 1.202 | 0.964 | 1.291 | 0.368 | 0 | 0.511 |
| S4 | 0.334 | 0.304 | 0.347 | 1.223 | 1.011 | 1.236 | 0.402 | 0 | 0.424 |
| S5 | 0.322 | 0.334 | 0.339 | 1.232 | 1.022 | 1.241 | 0.417 | 0 | 0.432 |

The Vickers hardness under load of 0.09807 (N)–HV 0.01–(ISO 6507-1) was measured on the surfaces of metallographic specimens—Figure 9.

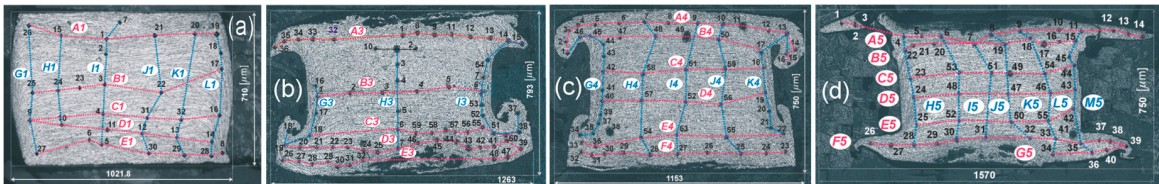

**Figure 9.** Metallographic cross-sections of samples: (**a**) Free upsetting; (**b**) UOM with a die CC; (**c**) UOM with a die FC; and (**d**) UOM with a die JC.

The hardness measures executed along the lines in Figure 9 are shown in Figure 10. The micro-part obtained in the UOM process is axially symmetrical. Starting from this assumption, the results of hardness measurements were expanded on the principle of mirror reflection in relation to the axis of the samples. The results in the form of micro-hardness distributions are shown in Figure 11.

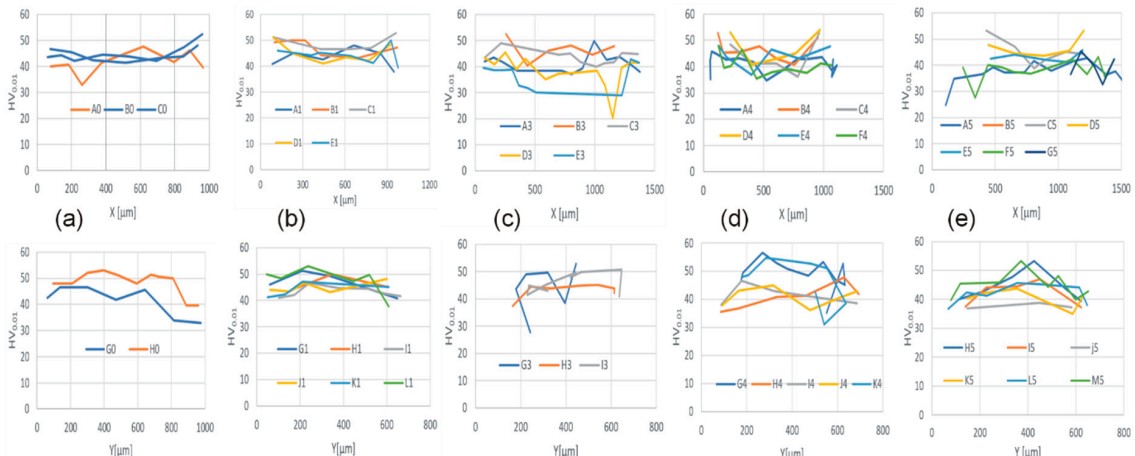

**Figure 10.** The results of micro-hardness measurements at the points marked in Figure 9 and in accordance with the courses of marked lines: (**a**) S0; (**b**) S1; (**c**) S3; (**d**) S4; (**e**) S5.

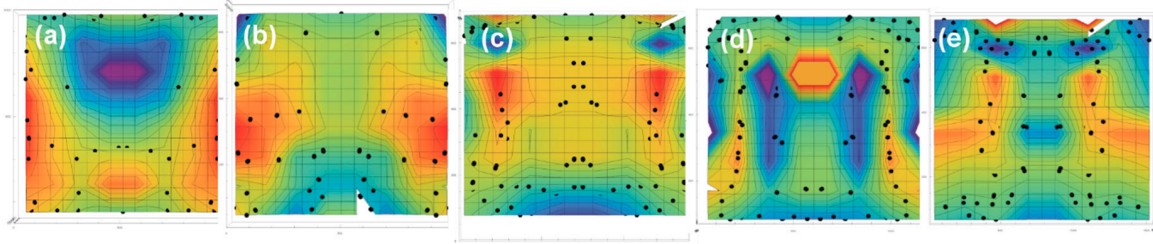

**Figure 11.** Micro-hardness distribution: (**a**) S0; (**b**) S1; (**c**) S3; (**d**) S4; and (**e**) S5.

## 4. Discussions

The deformation In UOM process starts at its periphery that is rolling on the surface of the bottom of the die and the face of the punch (Figure 12a,b). In the upper and lower parts of the samples, achieved ranges of deformations are impossible to be obtained in the process of free upsetting of this material [38]. Figure 12c shows the similar sample to these used in this work after the ultrasonic assisted upsetting process—the sample has cracked under deformation of the equivalent strain about 0.3. Achieving much larger deformations in the UOM process is associated with an increase in temperature resulting in increased deformability of the material.

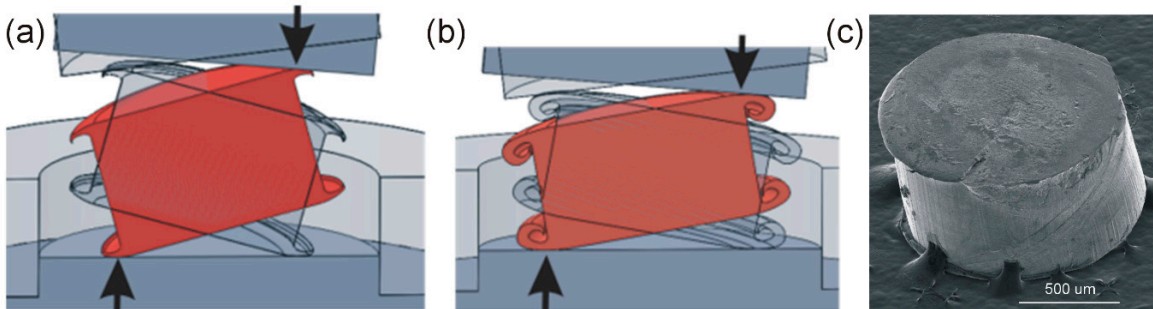

**Figure 12.** (**a**,**b**) The theoretical course of the process without the impact of die-cavity. (**c**) An example result of simple ultrasonic assisted micro-upsetting—previous investigation.

The UOM process creates a specific structure of the deformed sample. In the "classic" free upsetting, Figure 13a there are four zones: Z1 and Z2—the material is virtually unstrained; Z3—slight deformation and Z4—intense flow. These areas can be reflected in the microhardness distribution, Figure 11b. However, it should be remembered that the sample before deformation had the material structure after cold rolling, which was further changed in the blanking process.

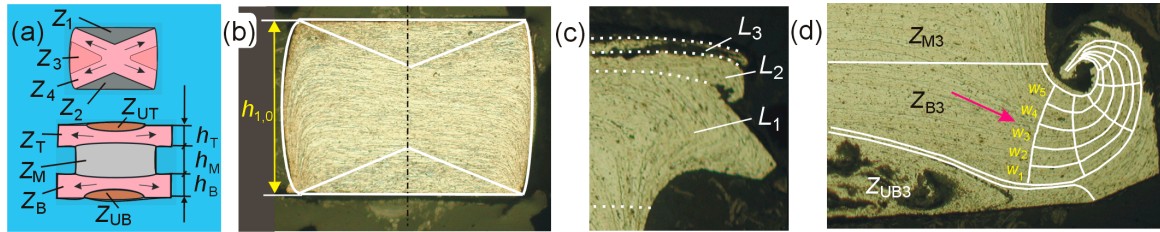

**Figure 13.** (**a**) Areas with diverse plastic features; (**b**) metallographic cross-section of S1 after simple upsetting; (**c**) close up q1of Figure 6a; close-up q2 of Figure 6a.

On the cross-section of samples deformed in the UOM process, five zones can be distinguished. They are shown in Figure 13d: $Z_M$-non-deformed material, $Z_T$ and $Z_B$ intense deformation, $Z_{UT}$ and $Z_{UB}$ material degradation zone under the influence of ultrasonic vibrations. In $Z_M$ zones, the courses of the central hardness lines (B in Figure 10c, C and D in Figure 10d, C and D in Figure 10e) are consistent

with the lines of the unstrained sample (B in Figure 10a ). The $Z_T$ and $Z_B$ zones are zones of intense material flow. The material in these zones flows in layers with different flow speeds. The shape of the flow line (Figure 13) suggests that the highest speed is at the contact surfaces with the tools. This speed as it moves away from the surface falls, causing the rim to bend as shown in detail $q_2$ of Figure 6a. This part of specimen is closer shown in Figure 13d, where the drawn flow lines form spirals due to the gradual drop in the flow velocity in subsequent layers from w1 to w5. Such processes can follow each other creating "layers overlapping". This is shown by the detail $q_1$ from Figure 6a. This area is closer shown in Figure 13c. Differences in the flow velocity of the material in the layers: $L_1$, $L_2$, and $L_3$ are probably related to the temperature distribution, which decreases as it moves away from the surface of the contact between the tools and the sample. The appearance of elevated temperature is caused by intense plastic deformation and presumably slipping of the edge of the samples on the tool surfaces that may occur in the initial phase of the rotation/rolling motion. Two phenomena act in this zone oppositely: strain hardening and temperature based softening as dynamic recovery and recrystallization. The first causes an increase in the hardness of the material and a second decrease in this hardness.

For this reason, in layers with a lower temperature - further from the contact surface, the hardness is higher in comparison to layers which are closer to the contact surface: compare curve C3 with curve D3 in Figure 10c and curve E5 with curve F5 in Figure 10e. The ZUT and ZUB zones are characterized by the occurrence of rows and voids. At the border of these zones a decrease in hardness is observed, see curve E3 in Figure 10c, curve F4 in Figure 10d and curve A5 in Figure 10e. The above described zones can be recognized in the hardness distributions shown in Figure 11.

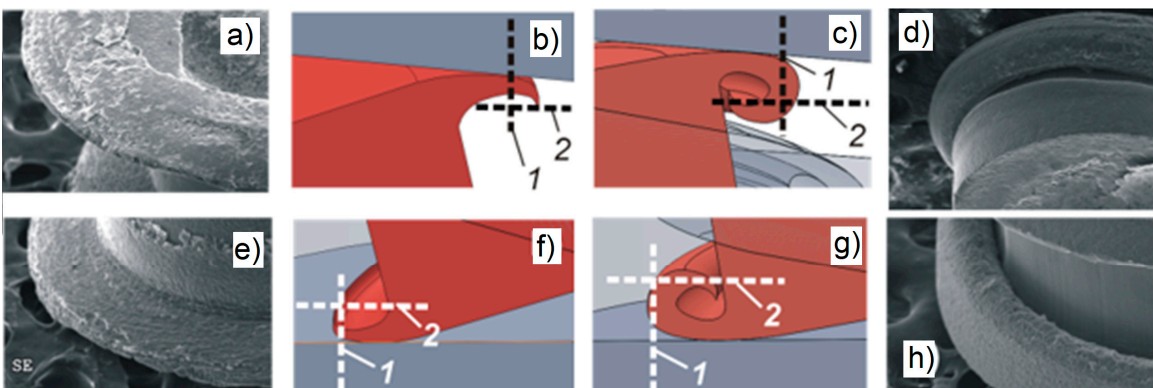

**Figure 14.** The course of the UOM process depending on the shape of die-cavity: (**a**,**b**) The theoretical course of the process without the impact of die-cavity; (**c**)–(**h**) The influence of vertical, 1, and horizontal, 2, limits.

The deformation proceeds according to the diagram shown in Figure 12a,b. The process runs in phases as evidenced by the forces in Figure 4b–d. Processes forces are very small, which is caused by a very long "distance of forming", small contact area and the so-called "dynamic effect". Forces are so small that their detection by the dynamometer of the testing machine is pointless. An exemplary record regarding the JC-S5 process, Figure 4f, is more a record of the resistances of the ultrasonic head guiding system than process force.

Along with the UOM process the phenomenon of folding of the deformed edge caused by the difference in temperature and flow velocity of the deformed layers is intensified by increasing the angle of inclination of the rotating sample with its deformation. The edge of the sample is successively "bended" in subsequent phases of the process (Figure 14c). The use of shaped dies (Figure 2) caused the material flow to be modified. The deformation is modified by two kinds of possible limitations: Vertical limits—Figure 14d, h and horizontal limits—Figure 14a,e. The shape of die cavity refers to the

shape of the final sample only in the case of the FC process, see Figures 5c and 15b. In this case, the bottom diameter was calibrated, resulting in an increase in the force of the process (Figure 4c).

The probably presence of hydrostatic pressure in the bottom part of specimen results in the reduction of $Z_{UT}$ and $Z_{UB}$ zones by evolution of interface surfaces—white lines m and r in Figure 6b has been evolved into red lines n and s. In other cases, as shown in Figure 15 the final outline of the sample is different from the outline of the cavity of the die.

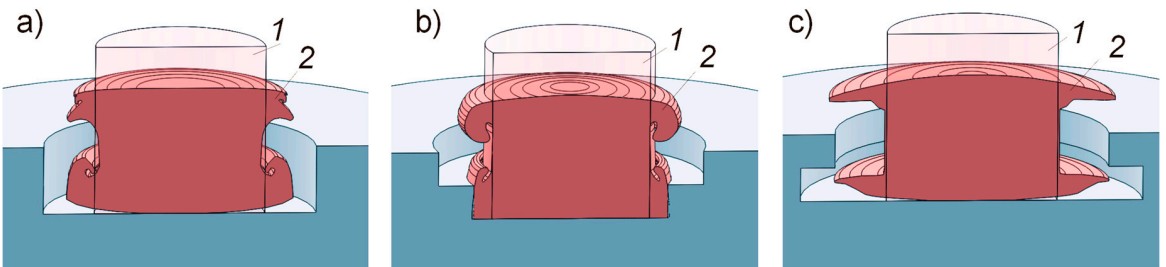

**Figure 15.** Initial, 1, and final, 2, shape of the workpiece in the die cavity in the processes: (**a**) CC; (**b**) FC; and (**c**) JF.

## 5. Conclusions and Future Work Suggestions

In this study, vibrations of 20 kHz frequency and 16 µm longitudinal amplitude were applied to deform an aluminum cylindrical micro-specimen inside three variants of modular-die in order to investigate the material flow during ultrasonic orbital microforming. Based on the experimental results of the testing stand with oscillatory stress measurement as well as the surface analysis by metallography, SEM, and micro-hardness tests, it can be concluded that:

- The OMF process causes the initial deformation of the edge material of the rotating workpiece. In further phases of the process, the deformation expands on both sides towards the center of the object.
- In the UOM process under investigation, only the upper and lower part of the workpiece is deformed, occupying approximately 70% of the volume. The middle part of the sample remains undeformed.
- During the UOM process, the material flows in layers of different flow velocity, which is most probably caused by the generation of heat at the interface of the workpiece-tool due to intense plastic deformation and friction resulting from the sliding edge of the rotating sample on the tool surface during the initial centrifugation phase.
- It was observed that the deformation takes place in phases between which breaks occur, during which the deformation does not take place.
- The progressive deformation can be modified by the internal shape of the micro-die in which the UOM process takes place.
- The internal shape of the die in which the UOM process takes place affects the final shape of the microproduct.
- The shape of the microproduct obtained during UOM process does not accurately reflect the shape of the matrix in which this process takes place.

UOM is a process in which dynamic phenomena related to the inertia of a rotating billet are used. In addition, ultrasonic vibrations that affect the contact phenomena and structural changes are involved. There is a high speed of deformation as well as rolling and sliding friction, which raise the local and global temperature during the process. The consequence is a whole range of thermally activated structural changes. The process is very complicated by the mutual interaction of many physical phenomena. The current state of knowledge on this subject should be considered as the initial.

Activities in many directions, such as: a description of traffic dynamics, numerical modeling, and modifications of the parameters as well as vibrating frequency and amplitude of the ultrasonic system as billet shape, material type, and structure are planned. The "statistical approach" to the research, and thus the development of results for more experiments is planned in the second phase of analysis. As the first, research is planned to lead to a better understanding of the occurring phenomena.

**Funding:** This research received no external funding.

**Conflicts of Interest:** The authors declare no conflict of interest.

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
