# Peer review of "Material Flow in Ultrasonic Orbital Microforming"

_metals, doi:10.3390/met9040475_

Round 1
Reviewer 1 Report
The manuscript is the continuation of the published in Metals in 2018 (Ultrasonic Orbital Microforming—A New Possibility in the Forming of Microparts) and must be accepted. However, the manuscript needs a revision, because in some parts it is difficult to understand the sentences. I will just list some examples:
Lines 36-38
Ultrasonic-vibration can reduce the ECAP [33] and micro-extrusions forces when friction is almost eliminated [34], and help in Increasing temperature by ultrasonic-vibration may reduce the flow stress, [35].
Lines 53-54
Gently eccentric impacts - coming from the "attacked" and lying sample at the resonant frequency cause a small precession of the punch.
Lines 58-61
After some time, the sample falls out of the acceleration resonance and falls to the bottom of the die. It lies still, and above it is the space. If punch touch the workpiece than process is started again. The process takes place in several stages.
Line 71
Figure 1. Experimental stand: a) schemat stanowiska (in Polish?),
Lines 71-80
Lack of coherence between the numbers presented in the legend of the figure and those of the text
Lines 83-86
On the basis of the laser displacement transducer, The amplitude AUS of the ultrasonic vibrations on the surface of the booster and the face of the punch was on the basis of the laser displacement transducer Keyence LK-85 H008 was determined, see Table 1.
Author Response
The manuscript is the continuation of the published in Metals in 2018 (Ultrasonic Orbital Microforming—A New Possibility in the Forming of Microparts) and must be accepted. However, the manuscript needs a revision, because in some parts it is difficult to understand the sentences. I will just list some examples:
Dear Revuer, Thank you very much for your help. All comments contribute to the improvement of the quality of this paper.
Lines 36-38
Ultrasonic-vibration can reduce the ECAP [33] and micro-extrusions forces when friction is almost eliminated [34], and help in Increasing temperature by ultrasonic-vibration may reduce the flow stress, [35].
modified
Lines 53-54
Gently eccentric impacts - coming from the "attacked" and lying sample at the resonant frequency cause a small precession of the punch.
Changed because or recent results
Lines 58-61
After some time, the sample falls out of the acceleration resonance and falls to the bottom of the die. It lies still, and above it is the space. If punch touch the workpiece than process is started again. The process takes place in several stages.
Changed and also reduced
Line 71
Figure 1. Experimental stand: a) schemat stanowiska (in Polish?),
done
Lines 71-80
Lack of coherence between the numbers presented in the legend of the figure and those of the text
Lines 83-86
On the basis of the laser displacement transducer, The amplitude AUS of the ultrasonic vibrations on the surface of the booster and the face of the punch was on the basis of the laser displacement transducer Keyence LK-85 H008 was determined, see Table 1.
Modified

Reviewer 2 Report
The paper is about an interesting technology strongly voted in the micro world. Despite being a fairly well-known technology, the merit of the paper is to deepen some aspects of the process with considerable detail, entering into a detail of considerable technical interest trying to motivate in a supported way some defects that emerged from intense experimentation.
However, the problem found at the base of the paper lies in the author's non-accurate re-reading of the paper.
English form is not accurate and there are also several omissions and confusion in the management of the figures and references to the same in the text. Another aspect noted is the excessive redundancy, sometimes, in expressing concepts.
I, therefore, try to give some suggestions that could improve the formulation of the paper.
First of all, the introduction does not sufficiently clarify what kind of scientific contribution the author has provided with this activity described. This deficiency is even more evident in the introduction, where a clear reference to the purpose of the paper does not emerge anywhere.
line 21: replace "lover" with "lower";
lines 22-23: please review the English form. What do you mean with the expression "it is a microforming"? Please try to improve the description of interesting concepts taken from citations.
line 31: remove "the" from "on the one hand";
line 32: put a coma at the end of the sentence after barrier and at line 33 replace "possibility" with "opportunity". at line 34 replace "vibrations of various.." with "vibrations at various".
please review the whole sentence at lines 33-34, it is unclear;
at line 40, remove "the" before "microforming" and before "often", please;
please, clarify the expression at lines 64-65; why can only be carried out as a microforming process?
Figure 1: please insert a legend for the numbers in the picture and reduce the caption with a recall to the legend;
review the whole period at lines 76-83 considering a recall to the legend;
lines 84-85, it seems that some text is missing, and the whole sense of the sentence is not clear. please review;
line 100: review the call to the Figure 1 that is wrong;
lines 102-103: it could be better to introduce a legend for the figure numbers as explained above. At line 103 "part" must be plural, so add an "s";
lines 120-121: please review the call to Figure 4;
line 127: review the sentence, for example writing "in figure 5 are reported images of the samples...";
line 137: remove "cross-sections" should be singular;
line 148: where can I find an explanation about i=3,4,5? You can write: H8i)T,M,B (Top, Middle, Bottom) with i=1,2,3 (specimens of figure?);
at line 150 is written "they", who are you referring to? please review;
line 152: "On their basis" is a wrong English form, please correct it; what are "n" and "s" related to c(1) T,B,n and c(i)T,B,s ?
line 166 some text missing, please read carefully next time!!!
Table 3 must be split in table 3-1 (Calculated values for V and h) and 3-2 (Calculated values for hn, ds and epsilon);
line 189: HV0,01 should be explained or recall an ISO norm, please; I don't think that "microhardness" exist as a word;
line 194: the expression "the runs of the lines" is wrong, you can modify in "the hardness measures executed along the lines of Figure 9..." or something like this;
lines 205-206: how many "achieved"?
line 207: you call Figure 12c, but what about Figures 12a and b? there are no calls to them in the paragraph;
line 208: what is "logarithmic strain"?
lines 222-242: all this paragraph needs a full review. Lots of calls to Figures are incomplete or not clear. For example, I can't understand when the author says at line 230 "Figure 6a shown in Figure 13d". the same problem at line 231 "Fig. 6a shown in Fig. 13c". Another point is that you can't call to Fig. if in the whole text you use Figure.
at line 227 replace "flow velocities" with "speeds" and at line 231 "layer assemblies" with "layers overlapping".
at line 250: I can't understand the expression "incremental character of forming.... later on you can't use "registration of them..." but "detection of them...".
at lines 252-253 "of" is used too much, please review the whole sentence.
at line 258, please put the call to figures between some round bracket;
at line 259, please review the sentence "3c and Figure 5c". You make quite always the same problem when you refer to Figures. Pay attention otherwise you loose clearness;
line 261-262, review the sentence "results of diminishing zones.." with "results in reducing Zut and Zub zones by..."
The conclusions should be introduced before starting with a list. Usually, a reader expects, in the end, to receive some information on future developments too.
Author Response
The paper is about an interesting technology strongly voted in the micro world. Despite being a fairly well-known technology, the merit of the paper is to deepen some aspects of the process with considerable detail, entering into a detail of considerable technical interest trying to motivate in a supported way some defects that emerged from intense experimentation.
However, the problem found at the base of the paper lies in the author's non-accurate re-reading of the paper.
English form is not accurate and there are also several omissions and confusion in the management of the figures and references to the same in the text. Another aspect noted is the excessive redundancy, sometimes, in expressing concepts.
Dear Sir, How to subdue enthusiasm if the UOM process seems to be one of a kind. Please pay attention to the fact that when it starts it goes regardless of the ram movement!
I, therefore, try to give some suggestions that could improve the formulation of the paper.
Thank you very much for such a detailed analysis of the text and very valuable suggestions for changes. Everlasting big improve the quality of the article. Thank you again.
First of all, the introduction does not sufficiently clarify what kind of scientific contribution the author has provided with this activity described. This deficiency is even more evident in the introduction, where a clear reference to the purpose of the paper does not emerge anywhere.
Improved – I believe
line 21: replace "lover" with "lower";
? Sorry – done
lines 22-23: please review the English form. What do you mean with the expression "it is a microforming"? Please try to improve the description of interesting concepts taken from citations.
line 31: remove "the" from "on the one hand";
done
line 32: put a coma at the end of the sentence after barrier and at line 33 replace "possibility" with "opportunity". at line 34 replace "vibrations of various.." with "vibrations at various".
Thank you for correction - done
please review the whole sentence at lines 33-34, it is unclear;
at line 40, remove "the" before "microforming" and before "often", please;
done
please, clarify the expression at lines 64-65; why can only be carried out as a microforming process?
done
The full answer is a bit complicated. Proving this thesis would require an analysis of the theory of mechanical similarity. This could be the content of the next publication. At the current level of the analysis of the process, it seems that it would be difficult to construct a machine that would be able to “push” a larger body of which we deal with in a rotational motion - fast enough for a gyroscopic effect to occur.
Figure 1: please insert a legend for the numbers in the picture and reduce the caption with a recall to the legend;
done
review the whole period at lines 76-83 considering a recall to the legend;
done
lines 84-85, it seems that some text is missing, and the whole sense of the sentence is not clear. please review;
done
line 100: review the call to the Figure 1 that is wrong;
done
lines 102-103: it could be better to introduce a legend for the figure numbers as explained above.
done
At line 103 "part" must be plural, so add an "s";
done
lines 120-121: please review the call to Figure 4;
done
line 127: review the sentence, for example writing "in figure 5 are reported images of the samples...";
done
line 137: remove "cross-sections" should be singular;
done
line 148: where can I find an explanation about i=3,4,5? You can write: H8i)T,M,B (Top, Middle, Bottom) with i=1,2,3 (specimens of figure?);
done: i – specimen number ; specimen name Si, there were used i=0,1,3,4,5 only.
at line 150 is written "they", who are you referring to? please review;
done
line 152: "On their basis" is a wrong English form, please correct it; what are "n" and "s" related to c(1) T,B,n and c(i)T,B,s ?
done, legend in Fig. 7 is added ; Cs and Cn are names of cylinders
line 166 some text missing, please read carefully next time!!!
I have a feeling that all used quantities are shown in Fig.7.
Table 3 must be split in table 3-1 (Calculated values for V and h) and 3-2 (Calculated values for hn, ds and epsilon);
done
line 189: HV0,01 should be explained or recall an ISO norm, please; I don't think that "microhardness" exist as a word;
done
line 194: the expression "the runs of the lines" is wrong, you can modify in "the hardness measures executed along the lines of Figure 9..." or something like this;
done
lines 205-206: how many "achieved"?
to many ? - done
line 207: you call Figure 12c, but what about Figures 12a and b? there are no calls to them in the paragraph;
done
line 208: what is "logarithmic strain"?
replaced with “equivalent”
lines 222-242: all this paragraph needs a full review. Lots of calls to Figures are incomplete or not clear. For example, I can't understand when the author says at line 230 "Figure 6a shown in Figure 13d". the same problem at line 231 "Fig. 6a shown in Fig. 13c". Another point is that you can't call to Fig. if in the whole text you use Figure.
done
at line 227 replace "flow velocities" with "speeds" and at line 231 "layer assemblies" with "layers overlapping".
done
at line 250: I can't understand the expression "incremental character of forming.... later on you can't use "registration of them..." but "detection of them...".
done
at lines 252-253 "of" is used too much, please review the whole sentence.
done
at line 258, please put the call to figures between some round bracket;
done
at line 259, please review the sentence "3c and Figure 5c". You make quite always the same problem when you refer to Figures. Pay attention otherwise you loose clearness;
done
line 261-262, review the sentence "results of diminishing zones.." with "results in reducing Zut and Zub zones by..."
The conclusions should be introduced before starting with a list. Usually, a reader expects, in the end, to receive some information on future developments too.
done

Round 2
Reviewer 2 Report
the authors answered quite to all the observations made.
In any case, there is still something that needs to be fixed before publication.
In my opinion the part between lines 331 and 354 need to be imporved a bit. For example it is necessary to explain why you say C in Figure 10c, E in Figure 10e, and so on. What are you referring too with these letters C, E, F, G?
At line 150, remove the coma;
at line 157 it is not necessary to report the reference [1] because is in the Figure caption;
at line 158 you need "s" at "two-parameter" because it is plural;
at lines 349-350 replace "distant" with "far".
Conclusions: please try to introduce the list in a better way, such as "we can sum up the results obtained from the study presented..." or something like this. In the final part, instead, try to explain how you intend to modify the parameters! Do you want to put on an experimental plan for statistical analysis of the process?
Author Response
Dear Sir,
Thank you very much for your help in improving the quality of my paper. Your comments helped me also on the more general matter. They let me to better identify the problems and also realise how many „white spots” should be cleared in the future.
-------------------------------------------------------------------------------
The authors answered quite to all the observations made.
In any case, there is still something that needs to be fixed before publication.
In my opinion the part between lines 331 and 354 need to be imporved a bit. For example it is necessary to explain why you say C in Figure 10c, E in Figure 10e, and so on. What are you referring too with these letters C, E, F, G?
“For this reason, in layers with a lower temperature - further from the contact surface, the hardness is higher in comparison to layers which are closer to the contact surface: compare curve C3 with curve D3 in Figure 10c and curve E5 with curve F5 in Figure 10e. The ZUT and ZUB zones are characterized by the occurrence of rows and voids. At the border of these zones a decrease in hardness is observed, see curve E3 in Figure 10c, curve F4 in Figure 10d and curve A5 in Figure 10e.”.
At line 150, remove the coma;
at line 157 it is not necessary to report the reference [1] because is in the Figure caption;
done
at line 158 you need "s" at "two-parameter" because it is plural;
done also in the case of [38] (in Fig. 12)
at lines 349-350 replace "distant" with "far".
done
Conclusions: please try to introduce the list in a better way, such as "we can sum up the results obtained from the study presented..." or something like this. In the final part, instead, try to explain how you intend to modify the parameters! Do you want to put on an experimental plan for statistical analysis of the process?
“In this study, vibrations of 20 kHz frequency and 16 um longitudinal amplitude were applied to deform an aluminium cylindrical micro-specimen inside three variants of modular-die in order to investigate the material flow during the Ultrasonic Orbital Microforming. Based on the experimental results of the testing stand with oscillatory stress measurement as well as the surface analysis by metallography, SEM and micro-hardness test, it can be concluded that:”.
“Activities in many directions, such as: a description of traffic dynamics, numerical modelling and modifications of the parameters as well as vibrating frequency and amplitude of the ultrasonic system as billet shape, material type and structure are planned. The "statistical approach" to the research, and thus the development of results for more experiments is planned in the second phase of analysis. As the first, research is planned to lead to a better understanding of the occurring phenomena.”.
